# Olive Mill Wastes: A Source of Bioactive Molecules for Plant Growth and Protection against Pathogens

**DOI:** 10.3390/biology9120450

**Published:** 2020-12-06

**Authors:** Fabio Sciubba, Laura Chronopoulou, Daniele Pizzichini, Vincenzo Lionetti, Claudia Fontana, Rita Aromolo, Silvia Socciarelli, Loretta Gambelli, Barbara Bartolacci, Enrico Finotti, Anna Benedetti, Alfredo Miccheli, Ulderico Neri, Cleofe Palocci, Daniela Bellincampi

**Affiliations:** 1Department of Chemistry, Sapienza University of Rome, 00185 Rome, Italy; fabio.sciubba@uniroma1.it (F.S.); laura.chronopoulou@uniroma1.it (L.C.); 2NMR-Based Metabolomics Laboratory, Sapienza University of Rome, 00185 Rome, Italy; alfredo.miccheli@uniroma1.it; 3Bio-Products-Bio-Processes Laboratory, Division of Biotechnology and Agriculture, Department for Sustainability, ENEA, Casaccia Research Center, 00123 Rome, Italy; daniele.pizzichini@enea.it; 4Department of Biology and Biotechnology Charles Darwin, Sapienza University of Rome, 00185 Rome, Italy; vincenzo.lionetti@uniroma1.it; 5CREA-Council for Agricultural Research and Economics, Research Centre for Agriculture and Environment, 00184 Rome, Italy; claudia.fontana@crea.gov.it (C.F.); rita.aromolo@crea.gov.it (R.A.); silvia.socciarelli@crea.gov.it (S.S.); anna.Benedetti1956@libero.it (A.B.); ulderico.neri@crea.gov.it (U.N.); 6CREA—Council for Agricultural Research and Economics, Research Centre for Food and Nutrition, 00184 Rome, Italy; loretta.gambelli@crea.gov.it (L.G.); enrico.finotti@crea.gov.it (E.F.); 7Order of Agronomists and Forestry Doctors, Province of Viterbo, 01100 Viterbo, Italy; studiobartolacci.agr@gmail.com; 8Department of Environmental Biology, Sapienza University of Rome, 00185 Rome, Italy; 9CIABC, Sapienza University of Rome, 00185 Rome, Italy

**Keywords:** *Olea europaea* L., olive mill wastes, plant growth, plant nutrition, plant protection, phenols, oligosaccharides, bioactive molecules

## Abstract

**Simple Summary:**

Olive oil is the most common vegetable oil used for human nutrition, and its production represents a major economic sector in Mediterranean countries. The milling industry generates large amounts of liquid and solid residues, whose disposal is complicated and costly due to their polluting properties. However, olive mill waste (OMW) may also be seen as a source of valuable biomolecules including plant nutrients, anthocyanins, flavonoids, polysaccharides, and phenolic compounds. This review describes recent advances and multidisciplinary approaches in the identification and isolation of valuable natural OMW-derived bioactive molecules. Such natural compounds may be potentially used in numerous sustainable applications in agriculture such as fertilizers, biostimulants, and biopesticides in alternative to synthetic substances that have a negative impact on the environment and are harmful to human health.

**Abstract:**

Olive oil production generates high amounts of liquid and solid wastes. For a long time, such complex matrices were considered only as an environmental issue, due to their polluting properties. On the other hand, olive mill wastes (OMWs) exert a positive effect on plant growth when applied to soil due to the high content of organic matter and mineral nutrients. Moreover, OMWs also exhibit antimicrobial activity and protective properties against plant pathogens possibly due to the presence of bioactive molecules including phenols and polysaccharides. This review covers the recent advances made in the identification, isolation, and characterization of OMW-derived bioactive molecules able to influence important plant processes such as plant growth and defend against pathogens. Such studies are relevant from different points of view. First, basic research in plant biology may benefit from the isolation and characterization of new biomolecules to be potentially applied in crop growth and protection against diseases. Moreover, the valorization of waste materials is necessary for the development of a circular economy, which is foreseen to drive the future development of a more sustainable agriculture.

## 1. Introduction

Olive tree (*Olea europaea* L.) cultivation for olive oil production is one of the most ancient agricultural practices known by mankind. Olive oil is an important component of the Mediterranean diet known for its high nutritional properties and beneficial health effects. On average, 3 million tons of olive oil are produced around the world every year and 2 million tons of this production takes place in the European Union (EU) representing the first producer, exporter, and consumer of olive oil in the world. The Mediterranean area alone including Spain, Italy, Greece, and Portugal covers around 99% of the olive oil EU production [1]. 

Olive oil extraction includes washing of olive fruits, fruit crushing, malaxation to brake off the emulsion and facilitate coalescence, and finally oil separation and extraction.

Olive oil extraction processes have improved over time due to the intensification of oil production and to the modernization of the technology aimed to upgrade the quality of the final product. 

The milling industry generates in a short period, usually 4 months (October–January), large amounts of waste (olive mill wastes, OMWs). For instance, an estimated average volume of olive mill wastewater (OMWW) ranging from about 0.3 to 1.2 m^3^/tons of processed olives and an average quantity of solid residue ranging from 500 to 735 kg/tons of processed olives have been reported depending on the adopted extraction systems [2]. OMWs due to their acidity, high levels of biological oxygen demand (BOD) and chemical oxygen demand (COD) are characterized by a high polluting and phytotoxic degree [3]. On the other hand, OMWs are a source of valuable molecules including plant nutrients, anthocyanins, flavonoids, polysaccharides, and several phenolic compounds [4,5,6] with potential industrial applications as fertilizers, antioxidants, antifungal and antibacterial drugs, cytoprotective agents, gelling and stabilizing agents in food preservation [4,7]. Consequently, significant efforts have been devoted to the transition from OMW detoxification to its valorization by optimizing the recovery of high added-value bioactive compounds to be commercially reused. 

The increasing consumption of vegetables and the ongoing climate changes, with negative effects on crop production and plant diseases diffusion, requires the large utilization of stimulants, fertilizers, and pesticides to improve plant growth, crop yield, and phytopathogens control [8]. The future challenge for modern agriculture is to operate in a sustainable way reducing the over-application of synthetic fertilizers and pesticides that have a negative impact on the environment and on human health and high persistence in the ecosystems. As an alternative to synthetic chemicals, biostimulants and biopesticides are the best candidates for sustainable integrated crop productivity and pest management [9,10]. Bioactive molecules with growth promotion and antimicrobial effects, identified and characterized in OMW by-products, have stimulated many researchers to employ these compounds as biostimulants, biopesticides, and plant protectants for crop improvement. However, more extensive field research is required to evaluate their effects to solve serious plant diseases affecting commercially important crops with a sustainable, large-scale, agro-economical perspective.

This review summarizes recent advances and multidisciplinary approaches in the identification and possible agronomic exploitation of valuable natural OMW-derived bioactive compounds. The acquired knowledge could also lead to the discovery of new plant growth and disease resistance regulators. 

## 2. Olive oil Extractive Methods and Olive Mill Wastes 

From the classic traditional discontinuous system, modern mills have moved to the use of continuous cycle extraction processes including two-, three-phase decanter systems [3] and, more recently, multi-phase decanter (MPD) technology in which all the oil extraction steps take place automatically and in succession with higher efficiency and capacity of centrifuge-based extraction [11].

In the three-phase decanter system, warm water up to 50 L for 100 kg olive paste is added during malaxation to enhance oil extraction. At the end of the process, a large quantity of OMWWs (0.3–1.2 m^3^/tons of processed olives) and dried olive pomace (DOP) (about 580 kg/tons of processed olives; 55% moisture) are produced [2].

The two-phase decanter system requires lower water addition, thus smaller amounts of OMWWs (about 0.2–0.3 m^3^/tons of processed olives) are generated and a semi-solid waste, defined wet olive pomace (WOP) (about 740 kg/tons of processed olive; 62% moisture) is produced [2]. The WOP treatment is difficult due to the high moisture and high concentration in solids, lipids, carbohydrates, and polyphenols [12].

The MPD technology is a modern two-phase system performed without adding water during the process. The introduction of a pulp-kernel separation system produces a dehydrated kernel-enriched fraction and a novel by-product, named Patè olive cake (POC) recovered during and not after the milling process by an exclusively mechanical treatment. POC consists of a wet fraction composed by olive pulp, olive skin, and vegetative water. POC is rich in several bioactive molecules, more concentrated with respect to OP derived from three- or two-phase systems [13,14,15,16].

OMWW is an expensive waste to dispose of and a major environmental concern in olive producing countries due to its high pollutant charge. OMWW is a dark-brown liquid (pH 3–6), constituted by a stable emulsion of vegetative water, process water, olive oil residues, and fragments of olive pulp. The OMWW composition depends on the extraction system, processed fruits, and processing conditions. OMWW consists of water (83–94% *w*/*w*) and organic compounds (4–18% *w*/*w*) including sugars, polysaccharides, tannins, organic acids, phenolic compounds, and lipids [17]. Its great complexity and variability in chemical composition represents a limit to its direct use as a raw material for industrial purposes.

## 3. Active Molecules in OMW and Their Analytical Characterization 

In order to valorize wastes and introduce valuable by-products into the production cycle, it is of paramount importance to define their chemical-physical properties, as well as their chemical composition. For a long time, the most important parameters to define wastes were the chemical-physical ones, such as total solid, volatile solid, fixed solid, oil and grease content, polyphenol content, volatile phenol content, organic nitrogen, COD, and reducing sugar content. 

More recently, analytical platforms were employed to better characterize the OMW parameters and a greater emphasis was placed on the evaluation of the antioxidant properties of these matrices [18,19].

The most used methods in order to detect the antioxidant capacity of single or multiple molecules, are oxygen radical absorbance capacity [20], total radical trapping antioxidant parameter [21], Ferric reducing antioxidant power [22], antioxidant reaction with an organic cation radical [23], (diphenyl-1-picrythydrazyl) copper (II) reduction capacity [24], and crocin bleaching assay [25,26]. 

The natural evolution of such studies is the investigation of the composition of OMW in terms of phenols, polyphenols, and sugars with more powerful techniques such as Fourier Transform IR, Mass Spectroscopy, Nuclear Magnetic Resonance Spectroscopy, or a combination of these [4,18,27,28,29,30,31,32,33] (see Table 1).

The content of bioactive molecules in OMW is heavily influenced by agronomic factors, such as pedoclimatic conditions of olive groves, as well as olive variety, harvest period, production year, and extraction process, as well as microbial treatments [6,34,35]. This observation implies that the chemical characterization of OMW is a step which must be repeated every time a new batch is to be employed for any kind of application [36].

Only recently, POC is being considered for other uses in addition to biofuel production [15,16], and as such a more detailed biochemical analysis is necessary. HPLC chromatography and gas mass spectroscopy could be fast and economical methods to detect phenols [37], while hyphenated platforms such as ultra-performance liquid chromatography coupled with a mass spectrometry detector without pre-treatment of the sample [38] can provide greater selectivity and sensitivity. A summary of the most recent studies concerning the bioactive molecules of OMW is reported in Table 1, and some of the most important molecules and their properties are briefly introduced in the following paragraphs.

### 3.1. Phenols and Polyphenols

Phenolics in plants are mainly synthesized through the phenylpropanoid pathway. Several phenols and polyphenols have been detected in OMWs (e.g., 32.8–103.4 mg/g dry matter in OMWW and 3.0–10.6 mg/g dry matter in POC, see Table 1), the most important ones being tyrosol, hydroxytyrosol, and their secoiridoid derivatives oleuropein and ligstroside. These compounds play an important role against inflammatory, aging, cancer, bacterial, etc. [43]. From a technological point of view, hydroxytyrosol could be very useful, but its synthesis is very expensive and it is not commercially available in large amounts [44]. For such reasons, the possibility to recover these compounds from natural matrices such as OMW could be an important goal.

### 3.2. Secoiridoids

The most diffused secoiridoids in OMW (e.g., 18–91 mg/g dry matter in OMWW and 0.2–9.6 mg/g dry matter in POC, see Table 1) are derived from elenolic acid, which is often esterified with hydroxytyrosol or tyrosol to form the aglycons of, respectively, oleuropein and ligstroside [29,42,45,46,47]. Other common derivatives are based on decarboxymethyl dialdehyde elenolic acid (oleacein) instead of elenolic acid. These molecules possess several properties in addition to their antioxidant potential with oleuropein, ligstroside, and their aglycons showing a significant antifungal and antimicrobial activity, as well as potential health benefits [48,49,50,51].

### 3.3. Carbohydrates 

Carbohydrates in OMWs are released following cell wall degradation during the milling process and during the olives ripening process. The knowledge of the composition of the cell wall polysaccharides of the olive fruit and of OMWs is useful to evaluate their potential applications [52]. Green olives, mostly used for oil production, have a cell wall composed of 46% of pectin polysaccharides (mainly galacturonan and arabinans), 28% Cellulose, 17% Glucuronoxylan, 10% Xyloglucan, 1% Mannan, and Arabinose-Rich Glycoprotein in traces [53,54]. Differences can be revealed between cultivars and ripening stages. Pectin-related sugars (Galacturonic Acid; GalA, Rhamnose; Rha, and Arabinose; Ara) are abundant in olives early during ripening, whereas hemicellulose monosaccharides (Xylose; Xyl, Mannose; Man, Galactose; Gal, and Glucose; Glc) increase later during maturation [55,56].

Cell wall isolation from OMW typically requires a hot ethanol precipitation followed by an alkali, acid, or solvent treatment. The monosaccharide composition is usually performed by HPAEC-PAD or GS-MS analysis associated with several colorimetric assays [56,57,58]. Differently from the OP, where insoluble cellulose and hemicelluloses are mainly present, OMWW is enriched in soluble pectins [58,59]. Polysaccharides isolated from OMWW are mainly enriched in GalA, followed by Ara, Glc, and Gal [58,60]. Conversely, Nadour et al. found that Glc was the major monosaccharide in the cell wall from OMWW, followed by Rha, Gal, Ara, Man, Xyl, and GalA. GlcA and fucose (Fuc) appear as minor sugars [5]. The differences observed among different studies can be explained by the different extraction or analytical methods used, and/or by different varieties or ripening stages of the olives. Evidence suggests that OMW is a source of partially soluble cellooligo-saccharides (COS), pectooligosaccharides (POS), and xylooligosaccharides (XOS) [61]. These oligomers compose a class of value-added compounds with enormous potential. All these oligosaccharides play a fundamental role as plant growth promoters or as defenders against pathogens [62,63,64,65]. Moreover, cell wall-derived products may have different nutritional and physiological benefits [66,67]. Possibly due to a recent interest in these OMW-derived oligosaccharides, a fine characterization of their chemical structure is not yet available in the literature. Such information is essential for designing new paths for the exploitation of OMW and its by-products in agriculture. The presence of glycosylated phenolic acids was also proposed for OMWW [4,5,68].

OMW are also enriched in soluble sugars, including significant amounts of Glc and Man and small quantities of Sucrose and Fructose [69]. Mannose can increase tolerance to salt and osmotic stress as “compatible solute” and can play a role in the responses to pathogen attack, as well as being heavily used in the food industry [70,71]. However, caution is required since it can have a negative effect on plant growth and defense when administered to plants in high doses [72]. A polymeric mixture, named polymerin, was also recovered from OMWW. It is composed of polysaccharides (54.4%), melanin (26.1%), protein (10.4%), and minerals (11.06%) strongly linked through covalent and hydrogen bonds. Polymerin could be used as a potential amendment, and/or metal biointegrator and as a biofilter for toxic metals [73,74].

## 4. Sustainable Processes for the Isolation of Bioactive Molecules from OMWs 

Several extraction methodologies for the recovery of OMW-derived bioactive substances are mostly based on the use of organic solvents. However, there is an increasing need for green and sustainable extraction approaches at low environmental impact. Membrane-based techniques such as microfiltration (MF), ultrafiltration (UF), nanofiltration (NF), and reverse osmosis (RO), constitute one of the most effective approaches to separate, concentrate, and finally recover active compounds [75]. These technologies considered best available techniques (BAT) are characterized by high separation efficiency, easy scale-up, and high productivity. These features make cross-flow processes more performing than conventional separation technologies. The effectiveness of MF as a pre-treatment for OMWW clarification was investigated in coupling with the membrane distillation process. The MF permeate was further treated through a membrane distillation stage, in order to obtain both clean water and a fraction containing concentrated phenols and sugars for fertilization purposes [76]. With this strategy, hydroxytyrosol was the main phenolic substance recovered and concentrated to 8.16 g/L. The UF 50 kDa cut-off was applied to separate the phenolic fraction and to lower the COD content of OMWW [77]. Different operating conditions were evaluated in terms of rejection to: COD, color, total solids, and total phenolic content. The results indicate that the UF in acidic condition is a suitable pre-treatment for OMWW to improve phenolic compounds recovery and environmental impact reduction. Different UF and NF membranes were used to separate from OMWW a pectin and a phenol-enriched fraction. Specific UF membranes (25 and 100 kDa cut-off) were effective to separate pectins from a mixture of cations and phenols. Finally, NF was utilized to separate phenols from cations in the solution [78] making this approach useful to separate different molecules from OMWW. Oligosaccharides fractions (1–3 kDa) were separated from WOP treated in a hydrothermal reactor and subjected to further chemical and enzymatical hydrolysis. UF was applied to separate tetra-, tri- and di-galacturonic acids, neutral and acidic xylo-oligosaccharides, and low molecular weight oligosaccharides of xyloglucan [79]. A membrane process based on ceramic UF, polymeric NF, and RO was adopted on water extracts of OP in order to concentrate the target compounds. The UF concentrate contains polyphenols (550 mg/L) and carbohydrates (4000 mg/L), in comparison, the NF concentrate shows more polyphenols (652 mg/L) and less carbohydrates (3000 mg/L). RO retains the remaining organic fraction returning a clean permeate with a total COD of 284 mg/L [12].

Photocatalysis has been studied as the OMW pretreatment to be applied before membrane fractionation. In order to achieve economic feasibility of the process, the catalyst must be recovered but immobilized systems cannot be used due to the high opacity of OMWW. Magnetic core titania particles, recoverable by means of a magnetic trap, were developed [80]. Their use permitted efficiently performing the pretreatment process of the wastewater stream, using a suspended photocatalyst photoreactor, and recovering up to 98% of the catalyst. By adopting photocatalysis as a pretreatment step for membranes, it was possible to increase the process productivity of 19% on average. COD values below 1.3 g/L were measured in the final permeate streams, achieving the quality standards for irrigation.

Supercritical fluid extraction (SFE) is being considered an advantageous eco-sustainable technique for biomass fractionation, starting from solid dehydrated materials [81,82,83]. Olive leaves have been used as a polyphenols source for SFE. CO_2_ modified by water was more efficient than that modified by ethanol in extracting oleouropein from olive leaves [84,85]. Water swells the matrix, opening pores, and allowing better access to solutes. Other authors described the use of different waste materials such as pruning biomass, leaves, and exhaust pomace as a source of polyphenols [86]. For all materials, SFE afforded extracts with a higher concentration of total polyphenols and higher antioxidant activity compared to SE. Hydroxytyrosol was the most prominent detected compound. Schievano et al. proposed an integrated biorefinery concept for the management of pomace, using SFE for extracting polyphenols and fatty acids, followed by the thermochemical recovery of energy, biofuels, and materials [87].

## 5. Application of OWS and OMW-Derived Bioactive Molecules in Plant Growth and Protection 

### 5.1. Effects of OMW as Plant Biostimulants 

The new EU Regulation 2019/1009 [88] defines a plant biostimulant (PB) as: “An EU fertilizing product the function of which is to stimulate plant nutrition processes independently of the product’s nutrient content with the sole aim of improving one or more of the following characteristics of the plant or the plant rhizosphere: (i) nutrient use efficiency, (ii) tolerance to abiotic stress, (iii) quality traits, or (iv) availability of confined nutrients in the soil or rhizosphere”. 

The members of the European Biostimulant Industry Council also proposed general principles and guidelines for trials and assays to be performed to allow PBs to be placed on the EU market [89]. In the last 10 years (2011–2020), about 1000 scientific papers were published on PBs and OMWs are recognized as biostimulants, although there is still little specific research on them [90].

Palumbo’s research [91] indicates that humic acids extracted from an amendment obtained combining OMWs with a pre-treated organic material derived from solid urban waste can be used as PB in agriculture, thanks to their positive effects on biomass production, nutrition, and activity of enzymes implied in N metabolism and glycolysis. Other studies have shown that PB formed as a by-product of the two-stage process of squeezing olive oil can induce an increase in the protein content of maize grains up to 19% [9]. OMW, at low concentrations, can efficiently trigger positive metabolic and physiological responses in plants [9].

Phenols are important signaling molecules and it has been established that in adequate concentrations they can produce several positive effects in plants, even when they are exogenously applied or present in PB formulations [9,92]. Conversely, at concentrations as high as those normally recorded in OMW, phenols may be responsible for the inhibition of soil microbiome activity and induction of several phytotoxic effects, including reduced seed germination, plant growth impairment, and drops in productivity [93,94]. 

### 5.2. Effects of OMW on Soil Properties and Plant Nutrition

The effects of land spreading of OMW on soil properties have been investigated and the results show that the effects on soil properties and plant growth are different according to the kind of OMW (liquid or solid, raw or processed). The form and rate in which OMWs are applied to soils play a significant role in determining their effectiveness as organic fertilizers and in soil health [95]. On the other hand, many studies investigated the effects of unprocessed OMWW on soil characteristics, recording negative effects on soil properties [96] and phytotoxic effects in seeds and plants when OMW is used directly as an organic fertilizer [97]. These wastes, due to their acidity, high organic load, high levels of BOD, COD, presence of high and low molecular weight polyphenols, short and long-chain fatty acids, and inorganic substances, can be characterized by a high polluting and phytotoxic degree. The reduction of OMW toxicity has been related to the degradation of phenolic compounds considered as the main responsible for the toxic effects on seed germination, on bacteria and on different species of soil and aquatic invertebrates [98]. Nevertheless, OMWs characteristics make them suitable for use as low-cost soil fertilizers recycling the organic matter and mineral nutrients [99]. Moreover, the clarified OMWW can represent a convenient source of irrigation water in Mediterranean countries suffering from water scarcity [100,101,102,103,104,105]. To solve the environmental problems linked to OMW disposal costs and allow its application to agricultural soils, several physico-chemical and biotechnological processes have been proposed for the OMW treatment based on evaporation ponds, reverse osmosis, filtration, oxidation, thermal drying, aerobic and anaerobic treatments, composting, phyto-depuration, and phenolic components extraction [106]. The impact on soil properties depends on the techniques used. Treated OMWs have no toxic effects and in general enhance soil fertility and plant growth [102,107,108]. However, the technological process application is limited due to the high investment or running cost [109] and the most frequently used ways to dispose of OMW nowadays are the application to agricultural soils of unprocessed OMW or after composting [33,95,110,111,112,113] or co-composting [114,115,116].

From the literature, OMW, solid and liquid, raw or processed forms, may affect soil chemical, physical and biological properties in different ways that, in turn, influence the growth and yield of crops as follows.

-Chemical properties: The physico-chemical characteristics of raw or processed OMW are adequate for an agronomic use as an organic fertilizer such as a slightly acidic pH, a very high content of organic matter, and balanced concentrations of mineral elements [117]. An important advantage of OMW is that it is free of heavy metals and other potential pollutants [118]. Many studies reported the general increase of the organic matter, organic N, macro and micronutrients on the soil, in particular, the available K [99,109,119,120]. The long term application of OMW in general did not cause significant differences in pH, EC, P, Na. However, pH, EC, and salinity can increase temporarily in topsoil after spreading high rates (200 m^3^/(ha*year)) of OMWW [103,104,105,121,122,123].-Physical and hydrological properties: Different results are reported in the function of forms of OMW (solid or liquid), rates of application, and pretreatments.

Land applications of raw or composted PO increase water retention and saturated hydraulic conductivity, reduce bulk density, and enhance the stability of aggregates with improved water availability for the crops. The increase in water holding capacity and wilting point can be attributed to changes in soil structure, which result from increases in soil organic carbon [124,125,126].

However, the accumulation of salts coming from irrigation with OMWW could lead to the disintegration of the soil structure and therefore the decrease of the hydraulic conductivity and a temporary reduction in the soil infiltration rate. In the topsoil, irrigation with OMWW produced an increase of stability of aggregates, lower bulk density, and relatively higher total porosity, but lower macroporosity [104,127,128,129].

Further advantages of OMW application is the reduction of erosion, runoff, soil losses [124], and pesticides persistence and mobility, decreasing groundwater risk contamination [130].

-Biological properties: The high polyphenols content of OMW represents the most limiting factor for spreading on soils due to their antimicrobial and phytotoxic effects. Nevertheless, OMW polyphenols are rapidly degraded depending on environmental conditions [100]. In regards to the soil microflora, OMW exercises the following two contrasting actions: It stimulates the development of the microflora by temporarily enriching the soil in carbon and inhibits some microorganisms and phytopathogenic agents due to the presence of antimicrobial substances. Studies report that microbial counts increase with OMW quantities and frequency of spreading [119,131]. In particular, aerobic bacteria and fungi increase in proportion with OMW spreading rates. Furthermore, soil respiration [96,99] and soil enzyme activities (dehydrogenase, β-glucosidase, and urease) seem to be enhanced in OMW-amended soils [122,123].

A study on short and long-term effects of repeated OMWW applications [101] showed a temporary negative effect on microbial biomass carbon but also the ability of soil to restore normal values in the long term. Moreover, on treated soils, despite a reduction of arbuscular mycorrhiza fungal root colonization, an increased presence of arbuscules and vesicles was observed.

-Growth and yield of crops: Almost always positive responses on plant growth and yield performances are reported when treated OMW (by composting or co-composting) are used as a consequence of polyphenols biodegradation. However, fertilizations or irrigations with untreated OMW at high doses can harm seeds germination and have negative effects on plant growth due to the phytotoxic effects of the elevated load of polyphenols and high salinity. Recent articles concerning the effect of OMW on plant growth and yield performances are listed in Table 2.

### 5.3. Effects of OMW as Biopesticides in Plant Protection 

In the agronomic industry, a huge problem is represented by fungal and bacterial pathogens that cause significant crops losses and produce mycotoxins potentially harmful for human health.

Plant diseases are largely addressed using synthetic pesticides that have a negative impact on environment contamination, are harmful to health, and can also generate pest-resistance. Bio-pesticides have been recently described as the best candidates for the control of phytopathogens for a sustainable agriculture.

The application of OMWs by-products in crop protection against pests takes advantage of their antifungal and antimicrobial properties without negative effects on plant growth [135,136].

Several experimental evidences reported the fungicidal activity of OMWW against dangerous phytopathogens. 

The inhibitory activity of OMWW on in vitro mycelium growth of *Fusarium oxysporum*, *Pythium spp., Sclerotinia sclerotiorum*, *Verticillium dahlia,* and *Botrytis cinerea* was reported [137,138]. The strong fungicidal activity of water extracts of WPO, diluted 1:10, was demonstrated against *Phytophthora capsica* [139]. The filter sterilized OMWW also inhibited the in vitro mycelium growth of *B. tulipae*, *F. oxysporum*, *Aspergillus niger,* and *Penicillium* spp [140]. 

Furthermore, the OMWW fruit application strongly reduced *B. cinerea* mold formation in strawberries and red peppers, indicating a possible application of this by-product in fruit protection against post-harvest diseases [138].

The OMW sterilization abolished all these biological effects suggesting that the antifungal and protective effect on fruits and vegetables from post-harvest diseases were possibly due to thermolabile phenolic compounds, although a synergistic effect of phenols with other molecules could not be excluded. 

Increased accumulation of phenolic phytoalexins in plants can promote host defense against pathogens and the activity of phenols in plant defense against phytopathogens have been explored for their application as biopesticides in agriculture [141].

Due to their low solubility in the oil phase, only 2% of phenols are contained in olive oil. Phenols mainly occur in OP (~45%) and in OMWW (~53%) [135] by-products making these wastes an important source of these molecules [136]. 

Caffeic acid, protocatechuic acid, para-coumaric acid, ferulic acid, cinnamic acid, and oleuropein isolated from OP, at a concentration of 1000 ppm, exhibited fungicidal activity against numerous phytopathogens with a higher effect on *F. oxysporum* and *Verticillium* sp [142]. A powerful fungicidal effect of phenols extracted from DOP at a concentration of 0.1 and 0.2% (*w*/*v*) was also demonstrated against *Alternaria solani*, *F. culmorum, Phytophthora capsica,* and *B. cinerea* [143].

Evidences indicated that the antifungal effect of OMWW is related to low molecular weight phenols and in particular to hydroxytyrosol (HT) and tyrosol. OMWW and HT-rich extracts inhibited the growth of *Pseudomonas syringae* pv tomato and *Xanthomonas campestris* at concentrations of 72 and 40 g/L, respectively [144]. Interestingly, OMWW at a minimum quantity of 10 µL/mL added to the medium inhibited the in vitro growth of the devastating bacterium *Xylella fastidiosa*. Moreover, a bacteriostatic effect was exerted by specific mono and polyphenols associated with OWS such as cathecol and methyl cathecol (1 mM), caffeic acid (1 mM), oleuropein (10 mM), and verbascoside (1 mM) [145]. However, the mechanism for *X. fastidiosa* growth inhibition by phenols is still unknown [146,147].

To understand the molecular basis of HT antifungal activity, different chemically synthetized HT analogues were used against *A. flavus*, *A. fumigatus*, *F. oxysporum,* and mycelial growth. HT analogues altered the plasma membrane structure within some minutes after their addition to fungal cultures. The HT analogues effect was associated with the inhibition of transporter mediated xanthine uptake, indicating that the direct effect of HT analogues was the disruption of the fungal plasma membrane integrity and function [148].

The plant cell wall (CW) is a source of bioactive molecules [149]. Cell wall fragments, so called damage-associated molecular patterns (DAMPs), released after enzymatic degradation of CW polysaccharides can improve plant protection, as well as crop yield [150].

OMWW is a good source of polysaccharides. Cellulose, hemicellulose, and pectins are the main carbohydrates identified in olive mill by-products [5]. While insoluble cellulose and hemicelluloses were mainly found in OP, soluble pectic components were observed in OMWW at significant concentrations [58,67,151,152]. Long (polymerization degree DP:10–15) and short pectin fragments (DP:1–8) (oligogalacturonides; OGs) are the best characterized DAMPs and are effective as protectors against phytopathogens [153,154,155]. These danger signals have been shown to trigger the CW-mediated immunity system leading to the elicitation of defense responses and disease resistance, as well as improve crop yield. Peculiar OGs with prebiotic and antioxidant activity have been fractionated from OMWs [5,61,156,157]. 

Acidic xylooligosaccharides are antimicrobial active agents [158,159], neutral xyloglucans act as endogenous elicitors [159], and arabinoxylan oligosaccharides can trigger plant immune responses in crops [65]. OGs neutral and acidic xyloglucan oligosaccharides have been detected in olive fruits [58,151,152] and have been isolated and chemically characterized from WOP [79]. Although the role of these olive oligosaccharides in plant defense has not yet been explored, future research could reveal their potential properties as new DAMPs to be used in plant protection.

Recent articles concerning the effect of OMWs and derived bioactive molecules on phytopathogens are listed in Table 3.

It is known that microbial communities have the ability to colonize OMWs and OMW microbiota depends on different physico-chemical conditions of OMWs as well by the cultivation systems, the olive tree cultivar, and olive fruit harvesting practices [160]. The potential biotechnological and industrial applications of indigenous microbiota, isolated from OMWs, in suppressive properties against plant pathogens and OMW bioremediation and valorization are beyond the aims of this review. However, it is important to highlight that OWM microbiota could modify the pH of the olive waste substrates and the composition of bioactive molecules during the OMW conservation period, thus affecting their efficiency as biopesticide in plant protection. Consequently, in the analyses of the biological effects of the OMW-derived bioactive molecules, the effect of a possible microbial colonization, by indigenous microorganisms, needs to be considered and a strict control of the storage and working conditions is highly required, to correctly manage these wastes. 

## 6. Conclusions and Future Perspectives

While OMWs have been for a long time an environmental issue, nowadays they are a source of bioactive molecules to be used in agriculture as natural pesticides, biostimulants, or plant protectants in alternative to harmful agrochemicals. New types of wastes, in particular POC, must be thoroughly studied to identify all potentially useful components, such as oligosaccharides to be employed as plant protectanta, as well as phenols and secoiridoids with potential antimicrobial properties. The definition of molecular mechanisms of action, coupling and complementing the protection activity of phenol and carbohydrate fractions could represent an attractive scientific challenge. In particular, basic research in plant biology may benefit from the isolation and characterization of new biomolecules to be potentially applied in crop growth and protection against diseases. 

Researches on OMW as fertilizers have demonstrated two different potential uses: As biostimulants or as soil amendments. The phenolic contents in OMW composition, at adequate concentrations, is able to determine positive metabolic and physiological responses in plants. In addition to phenols, attention should be devoted to olive oligosaccharides for their potential role as elicitors of defense responses. Their characterization and effects as plant elicitors of defense responses have not yet been investigated. 

There are few studies aimed to investigate the specific and/or synergistic actions of phenols and oligosaccharides in the complexity of the soil-plant system. Further and more focused researches in these topics are needed as a challenge in the valorization of Olive Mill by-products in formulations complying to the new EU biostimulants regulations and towards a sustainable agriculture.

The OMW effectiveness against phytopathogens has already been investigated and confirmed in many scientific experiences although their recovery with green approaches and their possible use in field applications still need to be further explored. A step in this direction is the ABASA project (Agricultural By-products into valuable Assets for Sustainable Agriculture) recently founded by LazioInnova, Regione Lazio 2017–2020, which has the aim to characterize the phytochemical composition of POC and OMWW fractionated by membrane filtration technologies, as well as to isolate and assess the physiological role of bioactive molecules.

## Figures and Tables

**Table 1 biology-09-00450-t001:** Recent studies on the characterization of bioactive phenolic compounds in olive mill wastes (OMWs).

Molecule	Analytical Platform	Amount in OMWW (mg/g Dry Matter)	OMWW References	Amount in POC (mg/g Dry Matter)	POC References
Phenols					
Tyrosol	HPLC, MS, GC, NMR	1.0–2.8	[6,32,39,40,41,42]	0.4–1	[13,14,15,16]
Hydroxytyrosol	HPLC, MS, GC, NMR	0.9–24	[6,32,39,40,41,42]	0.6–2	[13,14,15,16]
Hydroxybenzoic acid	HPLC, MS, GC, NMR	2–9	[6,32,39]	/	/
Coumaric acid	HPLC, MS, GC, NMR	1–2	[32,39]	0.1–0.6	[13,16]
Gallic acid	HPLC, MS	2–6	[6,39]	/	/
Vanillic acid	HPLC, MS	0.1–0.6	[6,39]	0.5–0.8	[16]
Caffeic acid	HPLC, MS	0.3–1.9	[6,39]	0.9–5.0	[13,14,15]
Hydroxycinnamic acid	HPLC, MS	Detected	[41]	/	/
Polyphenols					
Quercetin-3-O-glucoside	HPLC, MS	0.5–2.1	[39,42]	/	/
Luteolin-7-O-glucosides	HPLC, MS	25–55	[6,40,42]	0.5–1.2	[13,15]
Secoiridoids					
Oleuropein	HPLC, MS, GC	18–92	[6,39,40,41,42]	0.1–9.2	[14,15,16]
Oleuropein aglycone	HPLC, MS	Detected	[40]	Detected	[14]
Ligstroside	HPLC, MS	Detected	[40]	/	/
Ligstroside aglycone	HPLC, MS	Detected	[40]	/	/
Olecantal	HPLC, MS	Detected	[40,42]	0.1–0.4	[13]

**Table 2 biology-09-00450-t002:** Recent studies on the effects of OMWs on plants growth and yield. * indicates the type of OMW and the plant growth performance

Plant Organism	OMW				Plant Growth Performance		Notes	References
	OMWW		OP		Positive	Negative		
	Raw	Treated	Raw	Treated				
Italian ryegrass		*			*		Germination index	[132]
Olive trees			* POC from MPD	* POC from MPD	*		Long-term field study	[110]
Maize	*				*		Field trials calcareous soil	[120]
Durum wheat Barley	*				*		Field trials	[103,105]
Olive trees	*				*		Long-term field study	[119]
Olive trees	*			*	*		Long-term field study	[133]
Faba bean	*				* dose 25 m^3^/ha		Pot trials with different doses	[97]
Olive trees	*				* Olive grove yield		Long-term field study	[122]
Grapevine	*				*		Long-term field study (11 years)	[99]
Olive plantlets			*			*	pot trials in greenhouse	[134]
Winter Weath	*	*			* (OMWW treated)	* (OMWW raw)	Field trials	[123]

**Table 3 biology-09-00450-t003:** Recent studies on the effects of OMWs and derived molecules on microbial phytopathogens.

Microbial Organism	Tested Sample	Notes	References
*Botrytis tulipae, Fusarium oxysporum, Aspergillus niger and Penicillium* spp.	OMWW	Reduction of in vitro mycelium growth Reduction of Scab-like lesions *B. tulipae* on infected tulip bulbs	[140]
*Xylella fastidiosa*	OMWW and OMWs-derived MF, UF, and NF fractions	Inhibition of in vitro bacterium growth at minimum quantity of 10 µL/mL	[145]
*Aspergillus flavus, Aspergillus. fumigatus, Fusarium oxysporum*	Hydroxtyrosol analogues	Severe inhibition of myceliar growth at 100 µM	[148]
*Xylella fastidiosa*	Cathecol, Methyl cathecol, Caffeic acid, Verbascoside, and Oleuperin	Inhibition activity due to a bacteriostatic effect of tested phenols at 1 and 10 mM	[145]

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
