# Peer review of "Olive Mill Wastes: A Source of Bioactive Molecules for Plant Growth and Protection against Pathogens"

_biology, 2020, doi:10.3390/biology9120450_

Round 1

Reviewer 1 Report

The review is well written, but some minor amendments are necessary:

Lines 72-73: Please indicate that the values of wastewater and solids are estimated; moreover, the reference [2] is not appropriate (please find new statistic data /reference) because the article indicates an OMWWs production of approximately 175,000 m3 from 209,000 tons of olives with a ratio of 0.84 m3/olive ton; so, considering an oil yield of 15% the ratio OMWWs / oil corresponds approximately to 5.60  m3/olive oil ton which do not fit with the ratio of 10 deriving from 30 million m3 of olive mill wastewater Mediterranean countries (lines 71-72) for 3 million tons of olive oil are produced around the world (lines 61-62).

It is absent a paragraph concerning the microbiological characteristics of the olive wastewater and that the OMWWs change their composition during a conservation over time because of the microorganism activities, so that analyses and evaluations are carried out on fresh/raw OMWW. In addition, some effects may depend on the microorganisms contained in the OMWWs and on pH if not controlled (e.g. paragraph 5.3).

In references a lot of specie names are not in Italics.

Finally, do you confirm that all 15 authors really contributed to the review?

Author Response

The review is well written, but some minor amendments are necessary:

Lines 72-73: Please indicate that the values of wastewater and solids are estimated; moreover, the reference [2] ( Khdair 2019) is not appropriate (please find new statistic data /reference) because the article indicates an OMWWs production of approximately 175,000 m3 from 209,000 tons of olives with a ratio of 0.84 m3/olive ton; so, considering an oil yield of 15% the ratio OMWWs / oil corresponds approximately to 5.60  m3/olive oil ton which do not fit with the ratio of 10 deriving from 30 million m3 of olive mill wastewater Mediterranean countries (lines 71-72) for 3 million tons of olive oil are produced around the world (lines 61-62).

The authors kindly thank the reviewer for this observation. The reference, previously indicated in the original manuscript, has been changed by adding a new one as well as  the new estimated value of different olive wastes, reported by the cited authors, are  included in the text (see Introduction paragraph of the revised version of the manuscript)

It is absent a paragraph concerning the microbiological characteristics of the olive wastewater and that the OMWWs change their composition during a conservation over time because of the microorganism activities, so that analyses and evaluations are carried out on fresh/raw OMWW. In addition, some effects may depend on the microorganisms contained in the OMWWs and on pH if not controlled (e.g. paragraph 5.3).

According to the reviewer’s comment, considerations on this topic have been added at the end of paragraph 5.3 of the revised version of the manuscript.

In references a lot of specie names are not in Italics.

As indicated the references have been now corrected, with the specie names formatted in italics and the atomic weights in superscript.

Finally, do you confirm that all 15 authors really contributed to the review?

We confirm that all the authors contributed to the review according to their specific expertise and as reported in the author contribution section.

Reviewer 2 Report

Sciuba et al. provide a review on the isolation and identification of olive mill waste (OMW) products that could be used in agriculture as an alternative to synthetic compounds. This is a very comprehensive review that will be useful to those working in the olive oil industry as well as to those interested in developing new natural health products from agricultural products.  The manuscript is very well written and has a useful level of detail, except that   Table information is not well used in the text.

Minor corrections:

74          define BOD and COD

98          multi-phase decanter abbreviation is MFD, not  DMF

101-104  define “quantity of OMWWs and   Dried Olive Pomace (DOP) since all other parameters are shown in numbers ( 55% moisture)

160,161 delete ‘degenerative’  ‘ for several purposes’

Table 1: Place the reference column at the end

Table 1 included in the main text???

 198 Define more clearly why “a fine characterization of the chemical structure is not yet available

Author Response

Sciubba et al. provide a review on the isolation and identification of olive mill waste (OMW) products that could be used in agriculture as an alternative to synthetic compounds. This is a very comprehensive review that will be useful to those working in the olive oil industry as well as to those interested in developing new natural health products from agricultural products. 

The manuscript is very well written and has a useful level of detail, except that Table information is not well used in the text.

Minor corrections:

74          define BOD and COD    

The acronyms BOD and COD have been defined in the text according to the reviewer’s indication (see paragraph 1 of the revised version).

98          multi-phase decanter abbreviation is MFD, not  DMF

The abbreviations have been corrected according to the reviewer’s indication (see paragraph 2 of the revised version).

101-104  define “quantity of OMWWs and   Dried Olive Pomace (DOP) since all other parameters are shown in numbers ( 55% moisture)

The data have been added in the text and the sentence, previously reported in the manuscript rewritten according to the reviewer’s indication (see paragraph 2 of the revised version).

160,161 delete ‘degenerative’  ‘ for several purposes’

The sentence was corrected according to the reviewer’s indication (see paragraph 3.1).

Table 1: Place the reference column at the end

The table was modified according to the reviewer’s suggestion (new version of Table 1, and see paragraph 3 of the revised version).

Table 1 included in the main text???

Information from Table 1 are included in text of the revised version of the manuscript according to the reviewer’s indication (see paragraphs 3.1 and 3.2 of the revised version)

 198 Define more clearly why “a fine characterization of the chemical structure is not yet available

The answer to the reviewer’s question is indicated in the text of the revised manuscript (see paragraph 3.3 of the revised version).

Reviewer 3 Report

The present work is pertinent.

Some minor corrections may be performed before final version, namely, and considering only the Introduction section:

  • Introduction: line #74 BOD and COD appear for the first time and have to be written in the extended form;
  • Introduction: Introduction: final part of item 1 repeats the same idea. Please reformulate;
  • Introduction: line #99 "... efficiency and capacity OF centrifuge-based extraction."

Author Response

The present work is pertinent.

Some minor corrections may be performed before final version, namely, and considering only the Introduction section:

Introduction: line #74 BOD and COD appear for the first time and have to be written in the extended form;

The acronyms BOD and COD are now defined in the text of the revised  manuscript (see paragraph 1 of the revised version).

Introduction: Introduction: final part of item 1 repeats the same idea. Please reformulate;

The final part of introduction was modified according to the reviewer’s indication (see Introduction final part of item1 of the revised version).

Introduction: line #99 "... efficiency and capacity OF centrifuge-based extraction."

The sentence was corrected (see paragraph 2 of the revised version).